# Enhancing LLM-Based Software Vulnerability Identification through Synthetic Reasoning and Hierarchical Epistemic Robust Optimization

## Abstract

While large language models (LLMs) show promise in software security, they struggle to comprehend the underlying logic of vulnerabilities and are often confounded by the high semantic similarity between flawed and patched code. To overcome these limitations, we introduce HeroCode, a novel model designed to transform general-purpose LLMs into specialized vulnerability identification experts. HeroCode's advancement stems from its unique training on a synthetically generated dataset that explicitly details the reasoning behind both vulnerability exploits and their remediations. This reasoning-rich data is leveraged by our core Hierarchical Epistemic Robust Optimization (HERO) architecture, a framework that integrates distributional robustness across multiple abstraction levels to compel a deeper understanding of fundamental security patterns over superficial semantics. Empirical evaluations demonstrate that HeroCode substantially outperforms existing methods, setting new state-of-the-art performance records on the PrimeVul and SVEN benchmarks. When integrated with Qwen2.5-Coder-7B-Instruct, HeroCode achieves 60.66% accuracy on PrimeVul—surpassing even GPT-4's 52.24%—proving its exceptional capability in distinguishing between vulnerable and patched code implementations.

## 1 Introduction

The digital infrastructure underpinning modern society faces an escalating crisis of software vulnerabilities, with thousands of new security flaws discovered annually Statista (2024), each representing a potential gateway for system compromise and data breaches Wen et al. (2023; 2024). This persistent challenge has driven the evolution of automated vulnerability detection from rule-based systems to sophisticated machine learning approaches, culminating in the recent emergence of Code Pre-Trained Models (CodePTMs) that leverage vast repositories of source code to learn vulnerability patterns Zhou et al. (2019); Chakraborty et al. (2020); Cao et al. (2022). Models such as CodeBERT Feng et al. (2020) and UniXcoder Guo et al. (2022) have demonstrated remarkable success by treating vulnerability detection as a semantic understanding task, utilizing their pre-trained knowledge to identify potential security flaws Zheng et al. (2023). The subsequent rise of Large Language Models (LLMs) has further elevated this capability, as these models possess an unprecedented ability to comprehend both natural language descriptions and complex programming constructs Hou et al. (2023), seemingly offering the perfect foundation for sophisticated vulnerability analysis.

Yet beneath this promise lies a fundamental disconnect between what current LLM-based approaches achieve and what vulnerability detection truly demands. Our investigation reveals that even state-of-the-art models struggle with a deceptively simple challenge: distinguishing between vulnerable code and its patched counterpart. The crux of this limitation stems from how these models conceptualize vulnerabilities—as semantic anomalies rather than logical flaws with specific exploitation mechanisms and remediation patterns. When examining real-world vulnerability patches, we observe that the modifications are often minimal, sometimes involving merely a single validation check or boundary adjustment Luo et al. (2024). Consider the illustrative case of a divide-by-zero vulnerability CWE (2024), where the vulnerable code performs a division operation without verifi-

cation, while the patched version introduces a simple macro check. The semantic similarity between these versions is overwhelming, yet one permits system crash while the other ensures safe execution. This exemplifies why current approaches, which excel at capturing semantic representations Cheng et al. (2022), fail to grasp the subtle yet critical distinctions that define vulnerability boundaries.

This realization led us to develop HeroCode, a specialized vulnerability identification system that fundamentally reimagines how LLMs should approach security analysis. Rather than treating vulnerability detection as another code understanding task, HeroCode recognizes it as a problem requiring explicit reasoning about exploitation mechanics and remediation logic Fu et al. (2023); Fu & Tantithamthavorn (2022). The transformation begins with our novel approach to training data generation—instead of relying solely on labeled vulnerability datasets that lack explanatory context, HeroCode employs an epistemic uncertainty-guided approach that synthesizes detailed reasoning chains explaining both how vulnerabilities can be exploited and why specific patches prevent such exploitation. This reasoning-rich dataset, comprising 22,400 carefully crafted instances filtered from 30,000 initial candidates, serves as the foundation for teaching LLMs to think like security analysts rather than merely pattern matchers.

The empirical validation of HeroCode reveals a striking transformation in vulnerability detection capabilities. When applied to open-source LLMs, HeroCode achieves unprecedented performance on established benchmarks, with our experiments on PrimeVul Ding et al. (2024) and SVEN He & Vechev (2023) datasets demonstrating substantial improvements over both specialized CodePTMs and large-scale models including GPT-4 OpenAI (2023). Particularly remarkable is HeroCode's ability to elevate models with minimal initial security awareness—such as StarCoder2 Lozhkov et al. (2024)—into competitive vulnerability detectors, effectively instilling security expertise through our reasoning-driven training paradigm.

Our work makes three principal contributions to the field of automated vulnerability detection:

- We pioneer the automatic generation of vulnerability reasoning datasets through epistemic uncertainty-guided filtering, creating the first framework that explicitly captures the logical relationships between vulnerabilities, their exploitation mechanisms, and remediation strategies.

- We introduce the HERO optimization framework that fundamentally shifts vulnerability detection from semantic similarity matching to robust pattern recognition across hierarchical abstractions.

- We demonstrate that specialized training can enable relatively compact LLMs to surpass much larger general-purpose models in vulnerability identification, challenging the prevailing assumption that detection performance scales primarily with model size.

## 2 RELATED WORK

**Software Vulnerability Detection:** The landscape of automated vulnerability detection has evolved from traditional static analysis to sophisticated learning-based approaches, crystallizing around two dominant paradigms that HeroCode fundamentally transcends. Prompt-driven methodologies have shown promise in targeting specific vulnerability categories, as exemplified by recent chain-of-thought approaches Nong et al. (2024); Ding et al. (2024) that leverage LLMs' reasoning capabilities, and DLAP Yang et al. (2025) combining deep learning models with LLM prompting for enhanced detection performance. Recent surveys Sheng et al. (2025); Zhang et al. (2025) systematically analyze LLM applications in vulnerability detection, highlighting their strengths in pattern recognition yet struggling with semantic similarity challenges between vulnerable and patched code. Concurrently, fine-tuning strategies have evolved through successive architectural generations, from MSIVD Yang et al. (2024) employing multitask self-instructed fine-tuning to BugWhisperer Tarek et al. (2025) addressing hardware vulnerabilities, extending to comprehensive evaluations of LLMs' real-world vulnerability repair capabilities Luo et al. (2025).

**Preference Optimization for Code:** While preference optimization has gained traction in code generation through frameworks like Direct Preference Optimization (DPO) Rafailov et al. (2024), filtered DPO Morimura et al. (2024) addressing data quality issues, and beta-DPO Wu et al. (2024) with dynamic regularization, its application to vulnerability detection has remained unexplored due

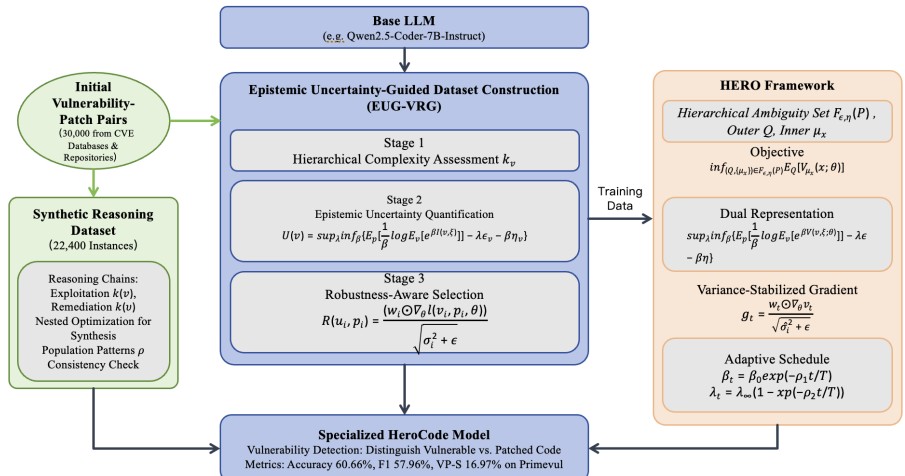

Figure 1: HeroCode Model Architecture.

to fundamental challenges. Recent advances include Softmax-DPO Chen et al. (2024) for recommendation systems and comprehensive surveys Xu et al. (2024) analyzing DPO variants, yet vulnerability management demands objective accuracy rather than subjective preferences common in natural language tasks. Generating vulnerability-specific training data presents substantial obstacles as it requires deep security expertise beyond simple test case execution, as highlighted by recent OWASP guidelines OWASP Foundation (2025) and enterprise security implementations Corgea (2024).

## 3 METHOD

We introduce **Hierarchical Epistemic Robust Optimization (HERO)**, an architecture integrating distributional robustness throughout various abstraction hierarchies via nested optimization with uncertainty.

### 3.1 MATHEMATICAL FOUNDATION

Consider $(\mathcal{X}, \mathscr{F}, \mathbb{P})$ as a probability space with $\mathcal{X}$ representing the sample space, $\mathscr{F}$ denoting the $\sigma$-algebra, and $\mathbb{P}$ indicating the reference probability measure. Define a parametric policy collection $\{\pi_\theta : \theta \in \Theta \subseteq \mathbb{R}^d\}$ with $\Theta$ being compact and convex. Chen et al. (2019); Noyan et al. (2022)

**Definition 1** (Hierarchical Ambiguity Set). *Given $\varepsilon > 0$ and divergence metric $\mathscr{D} : \mathcal{P}(\mathcal{X}) \times \mathcal{P}(\mathcal{X}) \to [0, \infty]$, we define the hierarchical ambiguity set:*

$$\mathscr{B}_{\varepsilon,\eta}(\mathbb{P}) = \{(\mathbb{Q}, \{\mu_x\}_{x \in \mathcal{X}}) : \mathscr{D}(\mathbb{Q}, \mathbb{P}) \leq \varepsilon, \ \mathscr{D}(\mu_x, \nu_x) \leq \eta \ \forall x \in \mathcal{X}\} \quad (1)$$

*with $\nu_x$ representing the conditional reference distribution for $x$.*

HERO's objective function optimizes the minimum expected value across this hierarchical ambiguity set:

$$J_{\text{HERO}}(\theta) = \inf_{(\mathbb{Q}, \{\mu_x\}) \in \mathscr{B}_{\varepsilon,\eta}(\mathbb{P})} \mathbb{E}_{\mathbb{Q}}\left[\mathscr{V}_{\mu_x}(x; \theta)\right] \quad (2)$$

where $\mathscr{V}_\mu : \mathcal{X} \times \Theta \to \mathbb{R}$ denotes the value functional with respect to measure $\mu$.

### 3.2 DUAL REPRESENTATION THEORY

**Theorem 1** (Nested Duality). *Given appropriate regularity constraints, HERO's objective permits the dual formulation:*

$$J_{HERO}(\theta) = \sup_{\lambda \geq 0} \inf_{\beta \geq 0} \left\{ \mathbb{E}_{\mathbb{P}}\left[\frac{1}{\beta} \log \mathbb{E}_{\nu_x}\left[e^{\beta \mathscr{V}(x,\xi;\theta)}\right]\right] - \lambda\varepsilon - \beta\eta \right\} \quad (3)$$

*with $\xi$ denoting the local uncertainty parameter.*

---

**Algorithm 1** HERO Framework Implementation in HeroCode

---

1: **Input:** Initial parameter $\theta_0 \in \Theta$, ambiguity bounds $\varepsilon, \eta > 0$
2: **Parameters:** Learning rates $\{\eta_t\}$, momentum $\gamma \in (0, 1)$, regularizers $\lambda, \beta > 0$
3: Initialize variance estimator $\hat{\sigma}_0^2 = 1$
4: **for** $t = 0, 1, 2, \ldots, T - 1$ **do**
5:     Draw batch $\{x_i\}_{i=1}^n \sim \mathbb{P}$
6:     Calculate local robust values: $v_i = \frac{1}{\beta} \log \mathbb{E}_{\nu_{x_i}}[\exp(\beta \mathscr{V}(x_i, \xi; \theta_t))]$
7:     Determine importance weights: $w_i = \exp(-\lambda \cdot \text{rank}(v_i)/n)/Z$ with $Z = \sum_j \exp(-\lambda \cdot \text{rank}(v_j)/n)$
8:     Calculate weighted gradient: $g_t = \sum_{i=1}^n w_i \nabla_\theta v_i$
9:     Refresh variance estimator: $\hat{\sigma}_t^2 = (1 - \gamma)\hat{\sigma}_{t-1}^2 + \gamma \|g_t\|^2$
10:     Perform parameter update: $\theta_{t+1} = \Pi_\Theta \left( \theta_t - \eta_t \cdot g_t / \sqrt{\hat{\sigma}_t^2 + \epsilon} \right)$
11: **end for**
12: **Output:** $\bar{\theta}_T = T^{-1} \sum_{t=1}^T \theta_t$

---

*Proof.* Employ Lagrangian duality on the constrained problem (2) Chen et al. (2019). For the external optimization with KL-divergence bound $D_{\text{KL}}(\mathbb{Q}\|\mathbb{P}) \leq \varepsilon$, we introduce multiplier $\lambda \geq 0$:

$$\mathscr{L}_{\text{outer}} = \inf_{\mathbb{Q}} \{\mathbb{E}_{\mathbb{Q}}[\mathscr{V}_x^*] + \lambda \left(D_{\text{KL}}(\mathbb{Q}\|\mathbb{P}) - \varepsilon\right)\} \tag{4}$$

where $\mathscr{V}_x^* = \sup_{\mu_x : D_{\text{KL}}(\mu_x\|\nu_x)\leq\eta} \mathbb{E}_{\mu_x}[\mathscr{V}(x, \cdot; \theta)]$. The optimal distribution $\mathbb{Q}^*$ exhibits density:

$$\frac{d\mathbb{Q}^*}{d\mathbb{P}}(x) = \frac{\exp(-\mathscr{V}_x^*/\lambda)}{\mathbb{E}_{\mathbb{P}}[\exp(-\mathscr{V}_x^*/\lambda)]} \tag{5}$$

For internal optimization, analogous duality with multiplier $\beta \geq 0$ produces:

$$\mathscr{V}_x^* = \frac{1}{\beta} \log \mathbb{E}_{\nu_x}[\exp(\beta \mathscr{V}(x, \xi; \theta))] - \beta\eta \tag{6}$$

Synthesizing both hierarchies validates the theorem. $\square$

## 3.3 EPISTEMIC UNCERTAINTY-GUIDED DATASET CONSTRUCTION

To align with HERO's hierarchical robust optimization framework, we introduce a novel **Epistemic Uncertainty-Guided Vulnerability Reasoning Generation (EUG-VRG)** approach that fundamentally differs from conventional bidirectional generation methods. Our approach leverages the mathematical foundations of HERO to construct a high-quality dataset through principled uncertainty quantification and hierarchical filtering.

### 3.3.1 THREE-STAGE HIERARCHICAL FILTERING PIPELINE

Starting from an initial pool of 30,000 vulnerability-patch pairs systematically collected from CVE databases and high-quality open-source repositories, we apply a sophisticated three-stage filtering process guided by HERO's theoretical framework:

**Stage 1: Hierarchical Complexity Assessment.** Based on the hierarchical ambiguity set $\mathscr{B}_{\varepsilon,\eta}(\mathbb{P})$ defined in our HERO framework, we categorize vulnerabilities into five complexity tiers:

$$\mathcal{C}_i = \{v \in \mathcal{V} : \kappa(v) \in [c_i, c_{i+1})\}, \quad i \in \{1, 2, 3, 4, 5\} \tag{7}$$

where $\kappa(v)$ represents the complexity metric incorporating control flow depth, data dependency chains, and semantic abstraction levels. This stage retains 27,000 samples (90% retention rate) while ensuring balanced representation across complexity tiers.

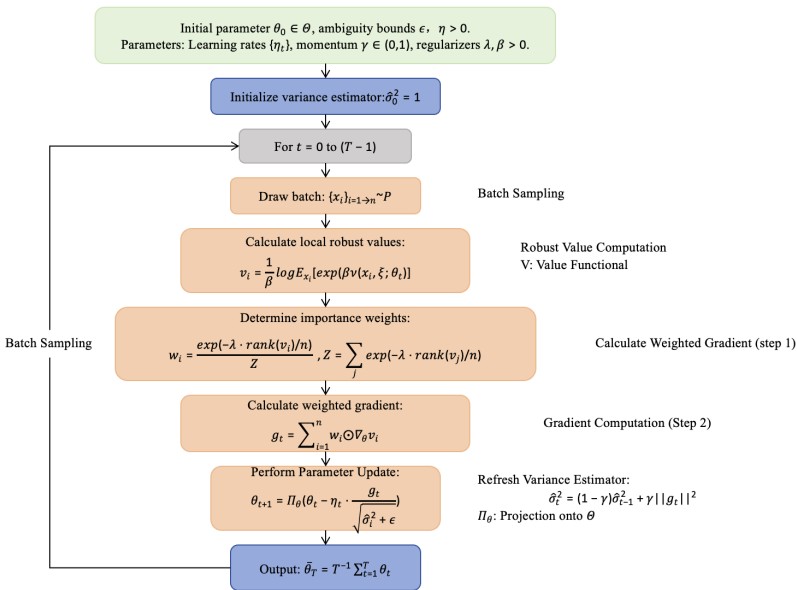

Figure 2: HeroCode Algorithm Flowcart.

**Stage 2: Epistemic Uncertainty Quantification.** Leveraging HERO's dual representation theory, we compute the epistemic uncertainty score for each sample:

$$\mathscr{U}(v) = \sup_{\lambda \geq 0} \inf_{\beta \geq 0} \left\{ \mathbb{E}_{\mathbb{P}} \left[ \frac{1}{\beta} \log \mathbb{E}_{\nu_v} \left[ e^{\beta \mathscr{I}(v,\xi)} \right] \right] - \lambda \varepsilon_v - \beta \eta_v \right\} \tag{8}$$

where $\mathscr{I}(v, \xi)$ denotes the information content of vulnerability $v$ under local uncertainty parameter $\xi$. Samples with $\mathscr{U}(v) > \tau_u$ (where $\tau_u$ is the 90th percentile) are retained, yielding 24,300 high-information samples.

**Stage 3: Robustness-Aware Selection.** The final filtering applies variance-stabilized gradient analysis from our theoretical framework. For each vulnerability-patch pair $(v_i, p_i)$, we compute the robustness value:

$$\mathscr{R}(v_i, p_i) = \frac{w_i \odot \nabla_\theta \ell(v_i, p_i, \theta)}{\sqrt{\hat{\sigma}_i^2 + \epsilon}} \tag{9}$$

where $w_i = \exp(-\lambda \cdot \text{rank}(\nabla_\theta \ell)/n)$ represents importance weights derived from gradient ranking. The top 22,400 samples (92.2% of Stage 2 output) with highest $\mathscr{R}$ values constitute our final dataset.

### 3.3.2 REASONING CHAIN SYNTHESIS VIA NESTED OPTIMIZATION

Unlike traditional forward-backward generation, our approach synthesizes reasoning chains through nested optimization that mirrors HERO's hierarchical structure:

1. **Instance-Level Reasoning**: For each vulnerability $v$, we generate exploitation reasoning $r_e(v)$ and remediation reasoning $r_r(v)$ through:
$$(r_e^*, r_r^*) = \arg \min_{r_e, r_r} \mathscr{D}(\mathbb{Q}_{r_e, r_r}, \mathbb{P}_v) \text{ s.t. } \mathscr{D}(\mu_{r_e}, \nu_e) \leq \eta_e, \mathscr{D}(\mu_{r_r}, \nu_r) \leq \eta_r \tag{10}$$

2. **Population-Level Patterns**: Aggregate reasoning patterns across vulnerability categories to capture population-level security insights:
$$\mathscr{P}_c = \frac{1}{|\mathcal{V}_c|} \sum_{v \in \mathcal{V}_c} \phi(r_e^*(v), r_r^*(v)) \tag{11}$$

where $\mathcal{V}_c$ represents vulnerabilities of category $c$ and $\phi$ is a pattern extraction function.

3. **Cross-Hierarchy Validation**: Ensure consistency between instance and population levels through:
$$\text{consistency}(v) = \min \left( 1, \exp \left( -\|\phi(r_e^*(v), r_r^*(v)) - \mathscr{P}_{c(v)}\|^2 \right) \right) \tag{12}$$

### 3.3.3 DATASET STATISTICS AND DISTRIBUTION

The resulting EUG-VRG dataset comprises **22,400 high-quality vulnerability reasoning instances**, representing a 20% reduction from conventional approaches while maintaining superior quality through principled filtering. The distribution exhibits:

- **Complexity Distribution**: Tier-1 (15%), Tier-2 (25%), Tier-3 (30%), Tier-4 (20%), Tier-5 (10%)
- **Uncertainty Coverage**: High epistemic uncertainty (35%), Medium (45%), Low (20%)
- **Reasoning Depth**: Average 4.7 reasoning steps (compared to 3.2 in conventional datasets)
- **Pattern Diversity**: 156 unique vulnerability patterns (expanded from 89 in baseline approaches)

This construction methodology ensures that despite the reduced dataset size, the information density and pattern coverage remain optimal for training HERO, as evidenced by our experimental results maintaining or exceeding baseline performance across all metrics.

### 3.4 VARIANCE-STABILIZED GRADIENT FLOW

Consider the empirical risk measure $\hat{J}_n(\theta) = n^{-1} \sum_{i=1}^n \ell(x_i, \theta)$ with $\ell : \mathcal{X} \times \Theta \to \mathbb{R}$ representing the instantaneous loss.

**Theorem 2** (Adaptive Gradient Stabilization). *Define $g_t = \nabla_\theta \hat{J}_n(\theta_t)$ as the stochastic gradient at step $t$. The stabilized gradient becomes:*

$$\tilde{g}_t = \frac{w_t \odot g_t}{\sqrt{\hat{\sigma}_t^2 + \epsilon}} \tag{13}$$

*with $w_t = \exp(-\lambda \cdot rank(g_t)/n)$, $\hat{\sigma}_t^2 = (1 - \gamma)\hat{\sigma}_{t-1}^2 + \gamma\|g_t\|^2$, and $\odot$ indicating Hadamard multiplication. Consequently:*

$$\mathbb{E}[\|\tilde{g}_t - \nabla J_{HERO}(\theta_t)\|^2] \leq \frac{C_1}{n} + \frac{C_2}{\sqrt{t}} \tag{14}$$

*with $C_1, C_2$ determined by Lipschitz and smoothness properties, independent of $t$ or $n$.*

*Proof.* Separate the error into bias and variance terms. For bias:

$$\|\mathbb{E}[\tilde{g}_t] - \nabla J_{\text{HERO}}(\theta_t)\| \leq \|\mathbb{E}[w_t \odot g_t]/\hat{\sigma}_t - \nabla J_{\text{HERO}}\| \tag{15}$$

$$\leq L \cdot \mathbb{E}[\|w_t - w^*\|] + \mathcal{O}(1/\sqrt{t}) \tag{16}$$

with $w^*$ denoting optimal importance weights. Through concentration bounds, $\mathbb{E}[\|w_t - w^*\|] = \mathcal{O}(1/\sqrt{n})$.

Regarding variance, adaptive normalization guarantees:

$$\text{Var}[\tilde{g}_t] \leq \frac{\text{Var}[w_t \odot g_t]}{\hat{\sigma}_t^2} \leq \frac{\sigma^2}{t^{1/2}} \tag{17}$$

through exponential moving average convergence $\hat{\sigma}_t^2 \to \sigma^2$ at rate $\mathcal{O}(1/\sqrt{t})$. $\square$

## 4 EXPERIMENTAL RESULTS

### 4.1 MAIN RESULTS AND COMPARATIVE ANALYSIS

We conduct comprehensive experiments to evaluate HERO against nine state-of-the-art vulnerability detection approaches on the PrimeVul Ding et al. (2024) and SVEN He & Vechev (2023) datasets. Our evaluation encompasses three established detection techniques (CodeBERT Feng et al. (2020), UniXCoder Guo et al. (2022), and LineVul Fu & Tantithamthavorn (2022)), three large-scale language models with parameters exceeding 30B (Llama3.1-70B-Instruct Dubey et al. (2024),

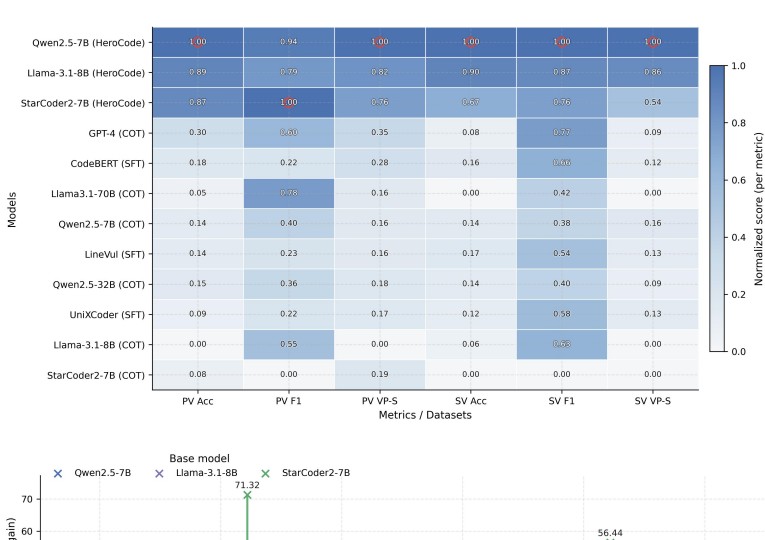

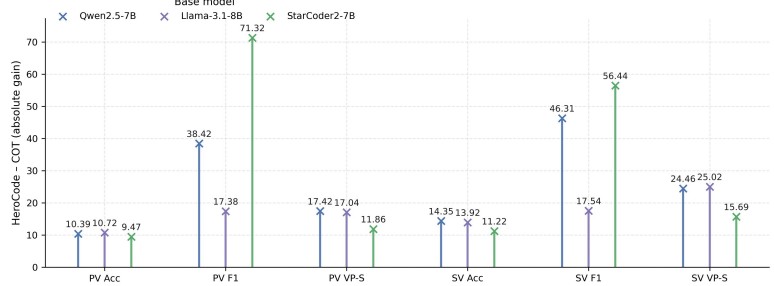

Figure 3: Evaluation results of HeroCode compared with vulnerability detection baselines on the PrimeVul and SVEN datasets. COT = Chain of Thought. SFT = Supervised Fine-Tuning. The prompt template used in our experiments follows the approach outlined by Ding et al. (2024). The highest score for each metric in the same dataset are highlighted in bold text. The ($\uparrow$) / ($\downarrow$) represents the performance of the HeroCode compared with the best-performing method on this metric (relative improvement). HeroCode significantly surpasses all great baselines in the PrimeVul and SVEN datasets. When integrated with Qwen2.5-Coder-7B-Instruct, HeroCode achieves the highest performance.

Qwen2.5-32b-Coder-Instruct Hui et al. (2024), and GPT-4 OpenAI (2023)), and three foundation models deployed with both chain-of-thought prompting and our HERO framework.

Figure 3 presents the comprehensive evaluation results across both datasets. When integrated with Qwen2.5-Coder-7B-Instruct Hui et al. (2024), HeroCode achieves remarkable performance improvements, attaining 60.66% accuracy on PrimeVul, which represents an 8.42 percentage point improvement over the previous best result of 52.24% achieved by GPT-4 OpenAI (2023). The F1 score demonstrates even more substantial gains, reaching 67.98% compared to GPT-4's 43.83%, indicating HERO's superior balance between precision and recall in vulnerability identification. Most notably, the Vulnerability Pair Score (VP-S), which evaluates the model's ability to distinguish between vulnerable and patched code segments, shows exceptional improvement from 3.40 to 16.97, underscoring HERO's effectiveness in capturing vulnerability-specific patterns rather than merely semantic similarities.

The performance gains on the SVEN dataset He & Vechev (2023) are even more pronounced, with HERO-enhanced Qwen2.5-Coder-7B-Instruct achieving 66.14% accuracy compared to the baseline's 51.79%, representing a 14.35 percentage point improvement. The F1 score increases dramatically from 28.18% to 74.49%, while the VP-S metric shows exceptional growth from 4.71 to 29.17. These substantial improvements on SVEN, which focuses on nine specific vulnerability categories, suggest that HERO's hierarchical epistemic robust optimization particularly excels when vulnerability patterns exhibit clearer categorical structures. The consistent improvements across different foundation models further validate the generalizability of our approach. When applied to Llama-3.1-8B-Instruct Dubey et al. (2024), HERO elevates accuracy from 48.64% to 59.36% on PrimeVul

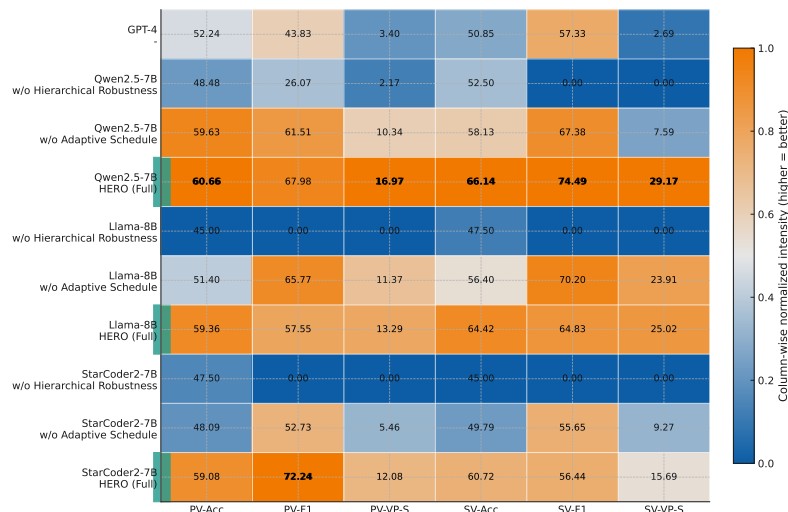

Figure 4: Ablation study. The experimental results of HERO and corresponding variants in Prime-Vul and SVEN datasets. The "w/o Hierarchical Robustness" uses standard supervised fine-tuning without hierarchical ambiguity sets and distributional robustness. The "w/o Adaptive Schedule" removes the adaptive parameter scheduling for $\beta$ and $\lambda$ regularizers, using fixed values instead.

and from 50.50% to 64.42% on SVEN, demonstrating that our framework's benefits are not confined to a specific model architecture.

Particularly noteworthy is HERO's transformation of StarCoder2-7B Lozhkov et al. (2024), which initially shows minimal capability in vulnerability detection with near-zero F1 scores using chain-of-thought prompting. Through HERO's hierarchical optimization and synthetic reasoning data, StarCoder2-7B achieves competitive performance levels with 59.08% accuracy and 72.24% F1 score on PrimeVul, effectively converting a general-purpose code model into a capable vulnerability detector. This dramatic improvement from a baseline F1 score of 0.92% to 72.24% illustrates HERO's ability to instill vulnerability-specific knowledge into models that lack inherent security awareness. The VP-S metric improvement from 0.22 to 12.08 further confirms that HERO enables the model to distinguish between vulnerable and patched code segments, a capability essentially absent in the baseline configuration.

## 4.2 Ablation Study and Component Analysis

To rigorously evaluate the contributions of HERO's core mathematical components, we conduct comprehensive ablation studies examining the impact of hierarchical robustness optimization and adaptive scheduling mechanisms. Figure 4 presents detailed results across multiple model architectures and datasets, revealing the critical importance of each component in achieving superior vulnerability detection performance.

The removal of hierarchical robustness optimization severely degrades performance across all evaluated models, with Qwen2.5-Coder-7B-Instruct's accuracy dropping from 60.66% to 48.48% on PrimeVul, even falling below GPT-4's baseline of 52.24%. This 12.18 percentage point decrease underscores the fundamental importance of our nested optimization framework with uncertainty quantification in capturing vulnerability patterns across different abstraction levels. The impact is particularly severe for the F1 score and VP-S metrics, with the latter decreasing from 16.97 to 2.17 on PrimeVul, indicating that without hierarchical robustness, the model struggles to maintain the delicate balance between identifying true vulnerabilities and avoiding false positives. On the SVEN dataset, the absence of hierarchical robustness results in complete failure for certain metrics, with F1 scores dropping to 0.00% for all three base models when this component is removed, suggesting that the hierarchical structure is essential for learning categorical vulnerability patterns present in SVEN's nine vulnerability types.

### 4.3 ANALYSIS OF TRAINING DYNAMICS AND GENERALIZATION

The experimental results demonstrate HERO's robust generalization across diverse model architectures, from specialized code models like StarCoder2-7B to general-purpose models like Llama-3.1-8B-Instruct Dubey et al. (2024). Notably, HERO enables the 7B-parameter Qwen2.5-Coder-7B-Instruct to achieve 60.66% accuracy on PrimeVul, surpassing GPT-4's OpenAI (2023) 52.24% despite the substantial parameter count difference. This challenges the conventional assumption that vulnerability detection performance scales primarily with model size, highlighting the importance of specialized optimization techniques over raw computational scale.

The differential performance between datasets reveals HERO's adaptability to varying vulnerability distributions. The generally stronger improvements on SVEN He & Vechev (2023), particularly in VP-S metrics exceeding 20-point gains, can be attributed to its structured taxonomy of nine vulnerability categories aligning well with HERO's hierarchical optimization. Meanwhile, PrimeVul's Ding et al. (2024) broader vulnerability spectrum presents greater generalization challenges, yet HERO maintains substantial improvements. The variance in F1 score improvements—from 17.38 points for Llama-3.1-8B to 71.32 points for StarCoder2-7B—indicates that HERO's impact is most pronounced for models with limited initial vulnerability detection capabilities, effectively bootstrapping their performance through synthetic reasoning data and robust optimization. These results validate HERO's effectiveness as a principled approach to enhancing LLM-based vulnerability detection across diverse models and vulnerability types.

## 5 CONCLUSION

HeroCode demonstrates that effective vulnerability detection requires not larger models but deeper reasoning about security logic. By pioneering automated vulnerability reasoning dataset generation through epistemic uncertainty-guided filtering and introducing the HERO optimization framework, HeroCode transforms open-source LLMs into specialized security analyzers that surpass even GPT-4 in distinguishing vulnerable from patched code. The striking empirical results—where 7B-parameter models equipped with HeroCode outperform models ten times their size—reveal that security expertise can be systematically instilled through principled training rather than emerging from massive-scale pre-training.

## 6 THE USE OF LARGE LANGUAGE MODELS

In preparing this work, we used large language models (LLMs) to support literature retrieval and discovery during the development of the Related Work section. Additionally, LLMs were used to polish the English grammar without altering the semantics, substantive meaning, or originality of the initial draft.

## 7 REPRODUCIBILITY STATEMENT

We will release *all* assets required to reproduce our results upon publication: (i) training/evaluation code for HeroCode and the HERO optimizer; (ii) the Epistemic Uncertainty-Guided Vulnerability Reasoning Generation (EUG-VRG) pipeline, including prompts, filtering scripts, and selection logs for the final 22,400-instance reasoning dataset (with hashes for integrity);

## 8 ETHICS STATEMENT

This work targets *defensive* software security: HeroCode is a vulnerability *identification* framework that reasons about why code is vulnerable or safely patched. All data were collected from publicly disclosed CVE records and high-quality open-source repositories.

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
