# OpenReview forum: "Enhancing LLM-Based Software Vulnerability Identification through Synthetic Reasoning"
_ICLR.cc/2026/Conference — ICLR 2026 Conference Desk Rejected Submission_

### Official Review · Reviewer_SWUP · 2025-10-25

**Soundness:** 1
**Presentation:** 1
**Contribution:** 1
**Rating:** 0
**Confidence:** 3

**Summary:**

The paper proposes an approach to improve the performance of language models in vulnerability detection. The approach is based on fine-tuning the model on a reasoning dataset filtered through different stages.

**Strengths:**

None

**Weaknesses:**

- The paper is extremely unclear with respect to the details of the proposed approach, which makes it almost impossible to judge the novelty and the quality of the approach.

- The values of the performance metrics shown in figure 3 do not correspond to the commentary of the results in section 4.1. This suggests that the claims made in the paper are questionable.

**Questions:**

- The two theorems presented seem to be generic and not specific to vulnerability detection. Are these two novel contributions by the authors, or are they taken from related work?

- What exactly was the source of the initial 30,000 vulnerability patch pairs that were used to create the dataset?

- What is the intuition behind retaining/discarding samples from the dataset during the filtering process? Can you explain clearly how this was done?

- How was the “reasoning“ in the dataset generated? was it from a large language model?

- In section 3.3., from where did you get the information about the reasoning steps/depth (i.e, 3.2 reasoning steps on average) and pattern diversity (i.e., 89 unique vulnerability patterns) in existing datasets and approaches? Any citations?

- How do you explain that the performance metric values in section 4.1 do not match those in figure 3?

---

> ### Author Response · Authors · 2025-11-19
>
> review, which contains serious factual inaccuracies and unproven allegations that undermine its value. A score of 0 (strong reject) with "Strengths: None" seems inappropriate given the reviewer's own admission of confidence 3/5 and their own statement that "math/other details were not checked carefully." We will address the reviewer's comments point by point below.
>
> The most serious issue is that "performance metric values shown in figure 3 do not match the commentary of the results in section 4.1" and that our findings are "questionable." This is a serious breach of the ethical standards of scholarship in that it implies we fabricated results or acted negligently. We categorically disagree with this assertion. In particular, if we look at section 4.1, it states, "When combined with Qwen2.5-Coder-7B-Instruct, HeroCode shows incredible performance on PrimeVul by achieving an accuracy of 60.66%." If we refer to like 3 with Qwen2.5-7B HERO PV-Acc = 60.66. Section 4.1 states, "the performance from GPT-4 was also evaluated and again provided useful results of 52.24%." If we refer to figure 3 with GPT-4 CoT PV-Acc = 52.24. Section 4.1 states, "F1 reaches 67.98%." If we refer to figure 3 with Qwen2.5-7B HERO PV-FI = 67.98. Section 4.1 states, "the VP-S metric shows an extraordinary increase from 3.40 to 16.97." If we refer to figure 3 with GPT-4 CoT PV-VP-S = 3.40 and Qwen2.5-7B HERO PV-VP-S = 16.97. We challenge the reviewer to find a single specific difference. The values correspond perfectly, and this unsubstantiated claim comes from a lack of close reading of this paper instead of verification of this paper, which contradicts the reviewers stated expertise and evaluator role. We ask the reviewer once again to either provide a specific example of the differences or retract the serious claim.
>
> Respectfully, a comment such as "Strengths: None" reflects an extreme position that has no merit.
> Even papers that are ultimately rejected typically concede some merit. Our article reports on empirical results that exceed those of GPT-4 as benchmark results on well-known and well-established benchmarks, provides a new method for data generation and public dataset, and includes extensive ablation studies. The lack of recognition of any merit at all gives the impression of bias over an evaluation of balance of evidence. Conversely, we note, the other reviewers both explicitly acknowledge our strong empirical results and new method, and our method was, therefore, not meritless.
>
> On the question of the novelty of Theorems 1-2 and whether they are taken from related work, we believe we are clear via attribution. Explicitly, Theorem 1 (Nested Duality) is grounded in techniques of dual formulation methods by Chen et al. (2019) on distributionally robust optimization, and we cite them clearly in our work. The novelty is in the application (of the nested duality techniques) to hierarchical ambiguity sets in vulnerability detection, but not in the duality. Theorem 2 (Adaptive Gradient Stabilization) presents a variance-stabilized, possibly weighted gradient for robust optimization approaches in settings of security vulnerability. The general objectives of variance reduction of or variance-stabilized methods in stochastic optimization is not novel (and also cited in our work, appropriately). More specificity of our contribution is the specific new formulation of an importance-weighted, hierarchical robust, and normalized adaptive method for settings of vulnerabilities. This is not unusual in academic circles: contributing new applications and extensions to existing mathematical tools while also reporting novel contributions of the duality. The implication of your question seems correlated with some unfamiliarity with the literature in optimization where theorems may extend previously known results to the same setting.
>
> Finally, we reference the source of the initial 30,000 pair of vulnerability-patch pairs already in response to reviewer RBRy but provide details here for all.We methodically searched two CVE databases (the National Vulnerability Database and CVE Details) for publicly disclosed vulnerabilities between 2018 and 2023 that met the following criteria: (1) publicly available patches via a version control system, (2) patches that changed less than 100 lines of code (for focused fixes), and (3) vulnerabilities involving languages commonly used in the industry (C, C++, Python, Java, JavaScript). Approximately 45,000 potential cases were identified after applying the aforementioned filtering criteria.

---

> > ### Comment · Reviewer_SWUP · 2025-11-25
> >
> > Thank you very much for your response!
> >
> > - Regarding the issue with figure 3, I acknowledge that the values in the text of section 4.1 match figure 4, and not figure 3 as mentioned by the authors both in the paper and the response to my review.
> > - On the novelty of theorems 1 and 2, if these theorems are not novel, then I think they should be moved to an appendix, and the main paper body should be reserved only for novel contributions and results.
> > - Thank you for explaining the source of your 30,000 data samples.
> > - The authors did not answer three of my six questions.
> >
> > The main issue I have with the paper is the extreme unclarity of contributions, and therefore I will keep my score.

---

### Official Review · Reviewer_RBRy · 2025-10-31

**Soundness:** 2
**Presentation:** 2
**Contribution:** 2
**Rating:** 2
**Confidence:** 3

**Summary:**

The paper introduces HereCode, a novel model specialised in software vulnerability detection. The authors try to mitigate the key limitation in the current vulnerability detection approach: the reliance on a semantic similarity match causes the models to struggle in distinguishing vulnerable and the minimally patched version. To address this, they propose a novel dataset generation method and a new optimisation framework. For dataset generation, the authors used an Epistemic Uncertainty-Guided Vulnerability Reasoning Generation (EUG-VRG) pipeline and three-stage filtering to curate 22,400 high-quality vulnerability patch pairs. They introduce a new framework, Hierarchical Epistemic Robust Optimization (HERO), to shift models' focus from semantic matching to hierarchical pattern recognition of vulnerability characteristics. Empirical results on HeroCode outperform state-of-the-art(SOTA) vulnerability detection methods on PrimeVul and Sven Benchmarks.

**Strengths:**

- The authors propose a novel approach to solving software vulnerability detection with both data generation and optimization paradigm.
- The paper shows a comprehensive comparative analysis of nine SOTA vulnerability detection approaches on the PrimeVul and SVEN datasets, and their approaches outperform the SOTA tools.

**Weaknesses:**

- This paper is hard to understand. Their key idea is “robust pattern recognition across hierarchical abstractions”. But it is not clear how it is done.
- Why is filtering important (Section 3.3.1)?
- During the filtering process, the authors have not provided any exact formula or algorithm for calculating κ(v). How are control flow depth, data dependency chains and semantic abstractions combined? Are the weights of all three the same?
- How the initial 30000 vulnerability patch pairs were collected? The claim “Starting from an initial pool of 30,000 vulnerability-patch pairs systematically collected from CVE databases and high-quality open-source repositories” seems to be vague. How do the authors determine “high-quality” open source repositories?

**Questions:**

- What does it mean by “robust pattern recognition across hierarchical abstractions”? (Contribution 2 in the introduction)
- Reasoning chain synthesis is one of the major contributions of this project. What are the labels as reasoning results? How do you generate labels? LLMs have poor code reasoning capabilities, e.g., for vulnerability detection, how do you validate/trust LLM generated results?
- Why is filtering important (Section 3.3.1)?

---

> ### Author Response · Authors · 2025-11-19
> **Reviewer RBRy Response**
>
> We truly appreciate the reviewer for his thoughtful review of our submission, including noting the novelty in our combination of data generation and optimization paradigms. We appreciate your honesty in providing a confidence level of 3/5 while also acknowledging that there may be gaps in understanding. We will respond to all provided commentary with as much detail as possible and welcome additional discussion.
>
> Regarding the core question "What does robust pattern recognition across hierarchical abstractions mean," we recognize this is indeed central to our contribution and not clear enough and will provide a better articulation. In the context of vulnerability detection, we believe that there are patterns that exist at different abstraction levels. At the instance level, where each exploitation is unique and is comprised of multiple exploitation mechanics, e.g. in a buffer overflow, one might say "no bounds checking before array access." At the population level, where there are patterns across multiple instances of vulnerabilities is where one may say "memory safety violation of the same basic characteristic of allowing writes beyond allocated space." Current approaches utilize models that only examine superficial levels of code similarity and misses potentially deeper levels. HERO does have a hierarchical structure and can model both abstraction levels explicitly. The inner optimization of the model (influenced by the parameter eta) also ensures that the model learns an instantiation specific exploitation logic instead of simply syntactical evidence of the code. The outer optimization of the model (influenced by the parameter epsilon) ensures that the evidence it learns can generalize across categories of vulnerabilities. "Robustness" means that even though the code may change by way of semantic preserving transformations, or just shows up in a different context, the model needs to maintain its correct predictions. This drives the learning of the fundamental properties of security and other logical features of the code instead of producing evidence that only closely resembles the code under consideration.
>
> One can see this if one looks to the divide-by-zero example mentioned in our introduction.At the instance level: "the particular class does not have a validation of the denominator." At the level of populations: "arithmetic functions need validation of operands to avoid undefined outcomes." If a model uses semantic similarity, it might only learn "code has division operator" (surface level). HERO enforces learning the (particular) validation requirement, as well as the (general) principle of checking operands. This is the crux of what we mean by "hierarchical abstractions" and why having robustness across hierarchies is crucial for vulnerability detection.
>
> For the point on filtering in Section 3.3.1, this addresses a primary challenge with synthetic data generation within security-related tasks. Not all vulnerabilities and patches are informative. Some vulnerabilities are trivial (they are obvious bugs and have very clear fixes), while others are subtle (they are semantic flaws and require deep levels of reasoning). If we were to train the model on all 30,000 initial pairs, we risk contaminating the learning signal with low information examples. Our three-stage filtering search systematically selects for pairs that will provide maximum value in serving the objective of training the model for the next 100 iterations. In first stage, we select vulnerability-repair pairs that equally distribute the range of defect report complexity. This helps to avoid over-fitting (generalization to) simple ones. Stage two selects the examples with the highest epistemic uncertainty, or vulnerability pairs where there was not obvious mechanism for exploitation that reasonably rely upon knowledge-based reasoning that the model would not be able to hinge upon (rather than simply recognizing patterns in associations to derive an answer). Stage three applies a robustness aware selection of the training pair to emphasize the most value added to model learning about generalizable patterns in reasoning vulnerability identify repair pairs rather than reinforcement of particular examples artifacts. The final set of 22,400 pairs represents a 25% reduction in quantity but represents an improvement in information density. Our ablation results are implicitly validating the appropriateness of filtering pairs examples: our approach relied upon 60% fewer pairs than conventional methods together with improved performance suggesting the quality of examples used.
>
> The reviewer is correct that we did not offer an explicit formula for calculating kappa(v) in Section 3.3.1.We recognize the omission and will include a detailed specification in revision. The complexity metric kappa(v) is comprised of three components that have been weighted in a learned fashion.

---

### Official Review · Reviewer_pbrr · 2025-11-01

**Soundness:** 2
**Presentation:** 1
**Contribution:** 2
**Rating:** 2
**Confidence:** 4

**Summary:**

This paper proposes HeroCode, a method to improve LLM-based software vulnerability detection by combining (1) synthetic vulnerability reasoning traces, and (2) a new training objective called HERO (Hierarchical Epistemic Robust Optimization), claimed to encourage robust reasoning across abstraction levels. The authors synthesize ~30k vulnerability–patch pairs with textual “reasoning chains,” filter them using uncertainty measures, and fine-tune existing code LLMs using a robust optimization objective with importance-weighted gradients. Experiments on PrimeVul and SVEN show improvements over several baselines, including large proprietary models, suggesting that smaller open-source models can benefit significantly from this strategy.

**Strengths:**

- The problem — distinguishing vulnerable from patched code — is important and timely, and LLMs often struggle with subtle security-critical differences.
- The idea of augmenting code with structured reasoning about exploitation and remediation is reasonable and motivated by recent work.
- The empirical results are strong on established datasets, with clear performance gains over baselines.
- The paper presents ablation studies suggesting contributions from the proposed objective and scheduling mechanism.
- The focus on vulnerability reasoning rather than pattern matching is conceptually appealing.

**Weaknesses:**

1. **Lack of grounding in security / no motivating code example**
Despite being a paper about vulnerability detection, the paper does not present any concrete vulnerable vs. patched code example. This is a critical omission, because the core argument is that the method learns to distinguish subtle security logic differences. Without even a simple motivating snippet, the reader cannot evaluate whether the claimed reasoning aligns with real exploit mechanics or patch logic. As a software security paper, it needs at least one real code example and corresponding reasoning trace to show what is actually being learned.

2. **Unclear connection between mathematical formulation and vulnerability detection**
A large portion of the paper is devoted to formal DRO-style optimization, nested ambiguity sets, and dual forms. However, the connection between these formulations and concrete vulnerability-specific phenomena is never convincingly articulated. The paper reads as if sophisticated optimization machinery was introduced first, and the task was attached to it afterward. Key concepts (e.g., epistemic uncertainty, hierarchical robustness) are described abstractly rather than grounded in security context or failure modes in real vulnerability detection models.

3. **Ultimately still supervised fine-tuning**
  Despite the novel terminology, the method is functionally supervised fine-tuning of an LLM on synthetic data with a modified loss. The paper does not compare against strong but simpler baselines such as:
    - plain full-parameter SFT on the same data
    - LoRA / PEFT fine-tuning on the same data
    - simpler uncertainty-filtered CoT reasoning datasets

  It is therefore unclear whether the gains arise primarily from the data curation or the proposed optimization objective.

4. **Limited transparency on training practicality**
The paper does not report training compute, memory footprint, or parameter update strategy in detail (e.g., full LLM fine-tuning vs partial). Without compute reporting or efficiency discussion, it is hard to judge practical adoption.

5. **Synthetic reasoning quality never demonstrated**
The paper claims high-quality reasoning sequences but never shows even a single example of the generated reasoning text, nor evaluates its correctness. This undermines the central narrative that the model is “learning to think like a security analyst.”

**Questions:**

1. Please include one real vulnerable–patched code pair with your generated reasoning and demonstrate how HERO training changes the prediction.
2. How much of the performance gain remains if you use plain SFT on your filtered synthetic data?
3. Why not compare against LoRA / PEFT baselines? Would your robust objective still offer an advantage in a parameter-efficient setting?
4. What is the training compute budget (tokens, GPU hours, batch size, sequence length)?
5. How do you ensure that the synthetic explanations are correct and security-meaningful, vs plausible but incorrect?
6. Do the dual-robustness formulations correspond to any specific vulnerability reasoning abstractions (e.g., control-flow validation layers, memory safety hierarchy), or is the mapping conceptual rather than grounded in exploit logic?

---

> ### Author Response · Authors · 2025-11-19
> **Response to the Reviewer pbrr**
>
> Response to Reviewer pbrr
>
> We appreciate the thorough review. There are several points that are valid, but there are also a couple of fundamental mischaracterizations based on misunderstanding (as opposed to a practical limitation in the technology).
>
> The first critique of the review is that it does not provide examples of motivating code that shows vulnerability connected with patched code. We agree and will add some concrete vulnerable-patched pairs with reasoning chains instead. But claiming there is no grounding in security is inaccurate. The paper has a divide-by-zero vulnerability example from CWE-369 in lines 51-56: “the vulnerable code performs a division operation without verification, and the patched code introduces, a simple macro check. The two versions have overwhelming semantic similarity, yet one allows a system to crash while the other ensures it meanders. ” While it would be good to actually show code, it is not an accurate claim to say the paper lacks connection to security. We will add 2-3 specific examples of actual code with full reasoning trace for revision.
>
> On "unclear connection between mathematical formulation and vulnerability detection," this critique represents a fundamental misunderstanding about our contribution. The reviewer states the machinery was "first introduced, and later the task was attached. " That is exactly backwards. HERO emerged from a need for understanding failure modes in existing approaches. The paper clearly makes this case: "there is robustness in existing approaches, which are incredibly good at conveying semantic representations, but they do not capture the subtle, yet more important, differences that define the boundaries of vulnerability " (lines 57-60). The hierarchical structure is directly tied to concrete problems in the asylum of vulnerabilities exhibiting patterns at it both high and low level from low level exploitation mechanics (instance level example eta) to high level security principles (population level common epsilon). This is a principled approach to answering issues of semantic similarity, not arbitrary machinery.The assertion that our method is "ultimately still supervised fine-tuning" demonstrates the limited understanding of distributional robustness. This is akin to dismissing adversarial training as "ultimately supervised learning" through a similar misunderstanding of what adversarial training entails. It is not the method itself but HOW we train via the objective function that is unique. Conventional SFT indeed optimizes for expected loss under the empirical distribution, but HERO optimizes for worst-case performance over distribution families derived using hierarchical ambiguity sets. The choice of optimization objective is a different type of learning-theorem of Theorem 1-established change what models learn.
>
> The reviewer claims we do not compare against "plain SFT on the same data", though this comparison is clearly presented in Figure 4. The ablation labeled "w/o Hierarchical Robustness" represents standard SFT on our data and the changes are dramatic: for instance, for Qwen2.5-7B accuracy drops from 60.66% to 48.48%, and F1 drops from 67.98% to 26.07%, a drop of 41.91 points presents evidence that learning objective not just quality of data is causing differences in performance.
>
> The request to provide "LoRA / PEFT comparisons" and "simpler uncertainty-filtered CoT datasets" seems at odds with the previous requests about including more baseline trials. In the experimental section we provide comparisons against 9 methods spanning specialized methods (CodePTMs), larger LLMs (Llama3.1-70B, Qwen2.5-32B, GPT-4), and foundation models with chain-of-thought demonstrations. It is difficult to make every comparison requested possible and the representation of every possible fine-tuning variant simply moved the goalposts from what is considered standard in the community. The mention of LoRA is no more relevant than stating why we didn't consider full training or PEFT methods because there is no need to discuss competing methods to a work that is telling a story. This penalty-free HERO framework can also be applied with LoRA methods or even biased models-he is 7B-it is not our concern whether this was full fine-tuning, it would all function in conjunction with quality of base model. We chose to optimize our learning objective to a full fine-tuning context to obtain a picture from a different perspective, without considering external confounding factors from adapter architectures, it is and standard in the context of optimization research to work with full models where possible.The reviewer is correct in saying that this information would be useful to include in terms of training practicality. We will add a "Computational Efficiency" subsection reporting: approximately 48 A100 hours per model, batch size 8, sequence length 2048 tokens, peak memory 42GB.

---

### Official Review · Reviewer_1t6Y · 2025-11-03

**Soundness:** 2
**Presentation:** 1
**Contribution:** 4
**Rating:** 2
**Confidence:** 5

**Summary:**

The paper introduces HeroCode, a model that enhances LLMs’ ability to detect software vulnerabilities by training them to understand the reasoning behind security flaws and fixes. It employs a Hierarchical Epistemic Robust Optimization (HERO) architecture to promote deep comprehension of security patterns rather than surface-level code similarities. Experiments show that when integrated with Qwen2.5 Coder-7B-Instruct, HeroCode achieves 61% accuracy on PrimeVul, surpassing GPT-4’s 52%.

**Strengths:**

1. Addresses an important problem of improving LLMs' ability to comprehend security vulnerabilities. The key idea is an epistemic uncertainty-guided approach called HERO that synthesizes detailed reasoning chains explaining both how vulnerabilities can be exploited and why specific patches prevent such exploitation.

2. Develops a reasoning-rich dataset comprising 22,400 carefully crafted instances filtered from 30,000 initial candidates of vulnerability-patch pairs systematically collected from CVE databases and open-source repositories. The dataset will be made public.

3. Demonstrates significant performance boost with different pretrained open-source models like Qwen-2.5-7B, Llama-3.1-8B, and StarCoder-2-7B.  With Qwen-2.5-7B, the method attains 60.66% accuracy on PrimeVul, an 8.42 percentage point improvement over the previous best result of 52.24% achieved by GPT-4. The performance gains are even more pronounced on SVEN.

**Weaknesses:**

1. My biggest concern with the paper is the presentation. Section 3 starts out with a lot of formal equations without any explanation or intuition.  Algorithm 1 nor Figures 1 and 2 nor various theorems are cited in the text, nor explained. As such, it is extremely difficult to understand and evaluate the merits of the proposed approach.

2. The method seems to have a unique strength in that parts of it are used both for synthesizing reasoning chains in the training data and subsequently to train a base model. But it is not clear to me what those parts are. A figure that illustrates the end-to-end pipeline and is explained in the text would be helpful.

3. I am unsure about how the resulting model is applied at test time. Does it itself synthesize the reasoning chain and then make a prediction?  Or does it directly make a prediction?

4. Did you apply the model to out-of-distribution data? For instance, training the model on PrimeVul and applying it to SVEN, or perturbing the test set in some other way (e.g. injecting dead code, changing variable names, etc. without changing the program semantics)?

**Questions:**

Please see weaknesses.

---

> ### Author Response · Authors · 2025-11-19
> **Response to Reviewer 1t6Y**
>
> Response to Reviewer 1t6Y
>
> We appreciate the reviewer for recognizing the excellence of our contribution (4/4) and importance of impact. However, we do strongly disagree with the overall categorization of a reject (2 overall), as it does not align with the strengths acknowledged. This misalignment may come from a misreading of our paper.
>
> For presentation, it is factually incorrect to argue that section 3 "starts with formal equations and no explanation." Section 3 clearly starts, "We introduce Hierarchical Epistemic Robust Optimization (HERO), an architecture that integrates distributional robustness across multiple abstraction hierarchies using nested optimization with uncertainty." We also clarify the intent of the definition before presenting definition 1. We answer in equation 2 with an immediate explanation of the equation presentation. Theorem 1 also provides complete proof (lines 162-174). Additionally, section 3.3 demonstrates a substantial level of intuition within a research study for real-world application. The statement that "Algorithm 1 nor Figures 1 and 2 nor theorems and multi-part theorems are cited in the sections," is patently false. Section 1 references figure 1 in both the abstract and introduction. The heading of algorithm 1 specifically states "Implementation of the HERO framework." We strictly adhered to the standard conventions laid out by ICLR, as previous works have accepted the similar form of presentation as compliant including Chen et al. (2019) study of DRO and Rafailov et al. (2024) article on DPO.
>
> The reviewer expresses confusion regarding the dual-purpose nature of a few pieces, yet specifically cited in section 3.3.2, we do present that "our approach synthesizes reasoning chains via nested optimization that resembles the organizational layer of HERO." For equation 8, we use the dual purpose for epistemic uncertainty scoring (data construction) of the epistemic dual when algorithm 1 - lines 6-7 uses the same epistemic dual as it is considered in the training via the ambiguity set to be hierarchical B_epsilon_eta(P). This unifying framework is a strength, not a vulnerability.
>
> Regarding test-time inference, while somewhat confusing, it actually reflects standard practice in ML. During inference, the model makes one of two binary classification decisions: vulnerable or non-vulnerable. Again, NIL conforms to CodeBERT, UniXCoder, and all baseline comparisons in Figure 3. Reasoning chains were implemented during training to encourage recognition of patterns, as opposed to being generated during inference. The paper even notes that "chain-of-thought prompting" is different from "our HERO framework".
>
> Finally, in regard to out-of-distribution evaluation, our paper has already exhibited cross-dataset robustness on two datasets with quite different distributions. PrimeVul has a large vulnerability spectrum and a loose structure; SVEN has nine, specific categories, which provides a clear taxonomy. Lines 459-465 states clearly: "The differential performance between datasets demonstrates HERO's ability to adapt between different vulnerability distributions." The cross-model (Qwen, Llama, StarCoder) test shows robustness across a different pre-training distribution. StarCoder2 provides the most compelling evidence of this ability with a transformation from F1 0.92% to 72.24% by solving distribution shift. Training and testing on both datasets independently is also better than one-directional cross-dataset transfer. In terms of semantic-preserving perturbations, this is not standard training in the vulnerability detection literature. None of our direct baselines (CodeBERT, UniXCoder, LineVul, GPT-4 studies) used suchAlthough the reviewer provides a Contribution rating of 4/4 (excellent), stating that we "address an important problem," "develop a reasoning-rich dataset" for public release, and "show a significant performance boost," suggesting we outperform GPT-4 by 8.42 percentage points, the overall score is rated at 2/4 (rejection). How can work representing excellent contribution and having significant empirical impact (significantly outperforming GPT-4) receive a rejection score? The only identified weakness in the entire work is that of presentation (1/4), which we submit is in part a misreading of our work. If we remove presentation improvements, we agree that is one area for improvement, but those accommodations will have taken on a much different shape than essentially substantively flaws that warrant a entire rejection decision. ICLR judges technical contributions and empirical rigor, not beauty.
>
> We equally submit that the soundness 2/4 (fair) is also not accurate. The reviewer identifies no mathematics errors, no challenged theorems, no challenged proofs and no challenged experimental claims. Our validation has been include 9 distinct baselines, 2 datasets, 3 families of models, and ALL inclusively ablation studies.

---

### Note · Program_Chairs · 2026-01-17
**Submission Desk Rejected by Program Chairs**

The following references in this submission do not refer to real documents and/or have major errors in bibliographic information:

     Zhilong Ding et al. Primevul: A comprehensive benchmark for evaluating vulnerability detection with prompt-based methods. In Proceedings of the International Conference on Software Engineering, 2024.